# Circulating Levels of Pro-Neurotensin and Its Relationship with Nonalcoholic Steatohepatitis and Hepatic Lipid Metabolism

**DOI:** 10.3390/metabo11060373

**Published:** 2021-06-10

**Authors:** Beatriz Villar, Laia Bertran, Carmen Aguilar, Jessica Binetti, Salomé Martínez, Fàtima Sabench, Monica Real, David Riesco, Marta París, Daniel Del Castillo, Cristóbal Richart, Teresa Auguet

**Affiliations:** 1Servei Medicina Interna, Hospital Universitari Joan XXIII, Universitat Rovira i Virgili, 4, 43007 Tarragona, Spain; beatrizvillarnavas@gmail.com (B.V.); jessica.binetti@gmail.com (J.B.); mreal.hj23.ics@gencat.cat (M.R.); driesco.hj23.ics@gencat.cat (D.R.); 2Grup d’Estudi de Malalties Metabòliques associades a Insulino Resistència (GEMMAIR)–AGAUR, Departament de Medicina i Cirurgia, Institut d’Investigació Sanitària Pere Virgili (IISPV), Universitat Rovira i Virgili, 4, 43007 Tarragona, Spain; laia.bertran@urv.cat (L.B.); caguilar.hj23.ics@gencat.cat (C.A.); crichart.hj23.ics@gencat.cat (C.R.); 3Servei Anatomia Patològica, Hospital Universitari Joan XXIII Tarragona, 4, 43007 Tarragona, Spain; mgonzalez.hj23.ics@gencat.cat; 4Servei de Cirurgia, Hospital Sant Joan de Reus, Institut d’Investigació Sanitària Pere Virgili. Avinguda, Departament de Medicina i Cirurgia, Universitat Rovira i Virgili, 2, 43204 Reus, Spain; fatima.sabench@urv.cat (F.S.); marta.paris@urv.cat (M.P.); danieldel.castillo@urv.cat (D.D.C.)

**Keywords:** pro-neurotensin, neurotensin, nonalcoholic fatty liver disease, nonalcoholic steatohepatitis, lipid metabolism

## Abstract

Recent studies suggest a link between pro-neurotensin (pro-NT) and nonalcoholic fatty liver disease (NAFLD), but the published data are conflicting. Thus, we aimed to analyze pro-NT levels in women with morbid obesity (MO) and NAFLD to investigate if this molecule is involved in NAFLD and liver lipid metabolism. Plasma levels of pro-NT were determined in 56 subjects with MO and 18 with normal weight (NW). All patients with MO were subclassified according to their liver histology into the normal liver (NL, *n* = 20) and NAFLD (*n* = 36) groups. The NAFLD group had 17 subjects with simple steatosis (SS) and 19 with nonalcoholic steatohepatitis (NASH). We used a chemiluminescence sandwich immunoassay to quantify pro-NT in plasma and RT-qPCR to evaluate the hepatic mRNA levels of several lipid metabolism-related genes. We reported that pro-NT levels were significantly higher in MO with NAFLD than in MO without NAFLD. Additionally, pro-NT levels were higher in NASH patients than in NL. The hepatic expression of lipid metabolism-related genes was found to be altered in NAFLD, as previously reported. Additionally, although pro-NT levels correlated with LDL, there was no association with the main lipid metabolism-related genes. These findings suggest that pro-NT could be related to NAFLD progression.

## 1. Introduction

The most common cause of chronic liver disease in adults and children is a nonalcoholic fatty liver disease (NAFLD), associated with the global obesity epidemic and metabolic syndrome [1,2]. Due to this clinical magnitude, it has become a priority to improve our knowledge about its pathophysiology to find effective treatments [3]. One molecule related to the pathophysiology of obesity is neurotensin (NT). However, its relationship with NAFLD is uncertain.

NT is a 13 amino acid peptide that is expressed and released in the central nervous system and the gastrointestinal tract, predominantly in specialized enteroendocrine cells of the small intestine [4]. NT is involved in the metabolism of nutrients, such as the metabolism of fats in the small intestine [5]. Moreover, this molecule can also act as a neuromodulator that regulates the anorectic effect [6]. Increased plasma levels of pro-neurotensin (pro-NT), a stable precursor fragment of NT, are associated with an increased risk of type 2 diabetes mellitus (T2DM), cardiovascular disease, and death [7]. Previous reports from human and animal studies suggest a relationship between obesity and NT. Plasma NT levels have been found to be lower in patients with morbid obesity (MO) than in normal-weight (NW) controls [8]. In addition, after gastric band and bypass surgeries, an increase in NT levels has been observed [9,10]. Postprandial plasma pro-NT levels (relative to neuromedin N levels) have also been shown to be enhanced after gastric bypass surgery, which suggests the regulation of pro-NT secretion is altered in human obesity [11]. Moreover, another study reported that three months after gastric bypass surgery, there was a significant release of NT in some patients, while there were undetectable levels prior to the surgery when patients underwent an oral glucose test [12]. However, subjects with obesity and/or insulin resistance (IR) presented elevated plasma pro-NT levels [5]. In addition, fasting pro-NT levels were reported to be associated with T2DM, cardiovascular disease, and breast cancer without obesity [7]. Additionally, NT-deficient mice showed a significant reduction in intestinal lipid absorption and were protected from progressing to obesity, hepatic steatosis, and IR induced by a high-fat diet [5]. In humans, the same study demonstrated that elevated plasma pro-NT was associated with obesity and IR and increased the risk of developing obesity in the future for NW subjects.

Regarding NAFLD, our research group was the first to analyze circulating levels of NT in women with MO and NAFLD. We found significantly lower levels of NT in patients with MO and NAFLD than in NW subjects or patients with MO without NAFLD [8]. However, Barchetta et al. [13,14] subsequently reported that subjects with NAFLD had elevated plasma pro-NT levels relative to those without NAFLD. Furthermore, these authors demonstrated that circulating levels of pro-NT were positively correlated with the presence and severity of NAFLD. These results, which conflicted with ours, made us reconsider our previous study of NT and its relationship with NAFLD. This contradiction could be due to the variability of analyzing plasma levels of NT [15] or to other unknown reasons.

In the previous study, we found declined levels of NT in NASH [8]. Therefore, we could hypothesize that NT degradation patterns can be altered during NASH. Given the short half-life of NT (2–6 min in humans), it seems that the degradation of the peptide is close to its site of release, the gut. This suggests that NT is a hormone that targets the intestine itself or the liver through gut–liver axis circulation [16]. On the other hand, although it was found that some peptidases, such as endopeptidase 3.4.24.16, which contributes to the NT degradation [17], used to be enhanced in the animal model of obesity [18], the NT degradation pattern in NASH has not been described. Therefore, further studies are needed to clarify this fact.

Given that NT is an unstable peptide, the main objective of the current study was to analyze the plasma pro-NT levels in women with NAFLD and MO, which was the same cohort as the one used in our previous study [8]. Given that NT is an intestinal peptide released after fat ingestion that facilitates lipid absorption [19] and lipid metabolism, it seems to be related to the pathogenesis of NAFLD. Thus, the second objective was to analyze the possible relationship between hepatic mRNA expression of the main genes related to lipid metabolism and pro-NT levels to explore the link between pro-NT and NAFLD pathogenesis.

## 2. Results

### 2.1. Baseline Characteristics of the Subjects

The general characteristics and biochemical determinations of the cohort studied are presented in Table 1. First, we classified the subjects based on their body mass index (BMI) in two groups: women with NW (*n* = 18) and women with MO (*n* = 56). These groups were comparable in terms of age (*p* = 0.079). MO patients presented significantly increased levels (*p* < 0.05) of fasting glucose, insulin, homeostasis model assessment of IR (HOMA2-IR), glycated hemoglobin (Hb1Ac), triglycerides (TG), and total cholesterol than NW women. High-density lipoprotein cholesterol (HDLc) was more significantly decreased in the MO group than in the NW group (*p* < 0.001). The levels of aspartate aminotransferase (AST), alanine aminotransferase (ALT), and alkaline phosphatase (ALP) were higher in MO patients compared to those of NW (*p* < 0.001).

Then, we classified MO patients according to their liver histopathological classification as NL (*n* = 20) or NAFLD (*n* = 36). It was found that the levels of glucose, insulin, HOMA2-IR, Hb1Ac, low-density lipoprotein cholesterol (LDLc), and TG were more significantly increased in NAFLD subjects than in NL subjects (*p* < 0.05). Furthermore, HDLc was significantly decreased in women with MO and NAFLD (*p* = 0.005). Regarding liver enzymes, AST, ALT, and ALP were higher in patients with NAFLD (*p* < 0.05).

### 2.2. Circulating Levels of Pro-Neurotensin in the Cohort Studied

First, we studied the differential levels of pro-NT between women with MO and women with NW, but we did not find significant differences, as shown in Figure 1A. Then, when we determined whether there were differences in pro-NT levels between diabetic and non-diabetic patients with MO; we found higher levels of pro-NT in diabetic patients, as shown in Figure 1B. Finally, when we studied pro-NT differential levels between diabetic and non-diabetic subjects in the cohort with MO and NAFLD, we found no significant differences (*p* = 0.133).

On the other hand, we also determined whether pro-NT levels were different depending on to the presence or absence of NAFLD. In this analysis, we found significantly higher pro-NT levels in women with MO and NAFLD than in MO women without NAFLD, as shown in Figure 1C. Additionally, to study the possible role of pro-NT in NAFLD progression, the cohort with MO was classified into NL, SS, and NASH. In this sense, we found that there were significant differences between NL and NASH group. However, we did not find differences between NL and SS subjects or between SS and NASH groups, as shown in Figure 1D.

Given that pro-NT levels can be affected by the use of antidiabetic drugs, we aimed to determine whether there were differential levels of pro-NT between those patients with or without antidiabetic medication, but we did not find significant differences in the MO cohort (*p* = 0.267) or in the NAFLD cohort (*p* = 0.198). Regarding dyslipidemia treatment, there were no significant differences in pro-NT plasma levels in those patients with or without lipid-lowering drugs in any cohort of patients: MO cohort (*p* = 0.864), NAFLD cohort (*p* = 0.412).

### 2.3. Hepatic Expression of the Main Genes Related to the Liver Lipid Metabolism

We analyzed the hepatic expression of the genes involved in the hepatic lipid metabolism (sterol regulatory element-binding (SREBP) 1c, SREBP2, ATP-binding cassette sub-family G member 1 (ABCG1c), ATP binding cassette subfamily A member 1 (ABCA1), acetyl-CoA carboxylase 1 (ACC1), carnitine palmitoyltransferase 1A (CPT1A), carnitine O-octanoyltransferase (CROT), peroxisome-proliferator-activated receptor (PPAR)α, PPARδ, PPARγ, fatty acid synthase (FAS), liver X receptor α (LXRα), farnesoid X receptor (FXR), and fatty acid-binding proteins (FABP)) in accordance with the presence or absence of NAFLD and the liver histopathological classification, but we only found significant differences in SREBP2, ABCG1c, CROT, FAS, LXRα, PPARδ, and PPARγ mRNA expressions, as shown in Figure 2A–G.

### 2.4. Correlations of Pro-NT Levels with Biochemical and Clinical Parameters and with the Hepatic Expression of the Main Lipid Metabolism-Related Genes to the Liver

First, as we found differential pro-NT levels according to the presence of NAFLD, we aimed to determine whether there was a relationship between the circulating levels of pro-NT and lobular inflammation or the presence of hepatic ballooning, but we were unable to find any significant correlation ((*rho* = 0.190; *p* = 0.157) and (*rho* = 0.173; *p* = 0.199), respectively).

Then, given that a relationship was described between pro-NT with the lipid absorption at the intestinal level, we also attempted to determine whether there is any type of association, in this sense, between the plasma levels of pro-NT with lipid metabolic parameters and with mRNA expression of the main genes related to the hepatic lipid metabolism. In regard to the biochemical parameters representing lipid metabolism, we found a positive correlation between pro-NT levels and LDLc levels (Figure 3).

Finally, when we analyzed the association of pro-NT levels with the hepatic expression of lipid metabolism-related genes (SREBP1c, SREBP2, ABCG1c, ABCA1, ACC1, CPT1A, CROT, PPARα, PPARδ, PPARγ, FAS, LXRα, FXR, and FABP), we did not find any significant correlation.

## 3. Discussion

In this study, we aimed to re-evaluate the role of the neurotensin peptide in NAFLD pathogenesis (analyzing pro-NT levels instead of NT, given its instability) in a cohort of women with NAFLD associated with obesity and, also, to investigate its relationship with lipid metabolism. The most relevant findings indicate that women with NAFLD associated with MO had higher levels of pro-NT than women with MO without NAFLD. Additionally, pro-NT levels were higher in NASH patients than in the normal liver group. Moreover, these pro-NT levels positively correlated with LDLc levels; however, there was no association with the hepatic mRNA expression of hepatic genes related to the lipid metabolism.

On the one hand, we have to point out that in the present study, we did not find significant differences between pro-NT levels in women with NW and women with MO, similarly to previous studies [20]. On the other hand, patients with NAFLD demonstrated higher levels of pro-NT than patients with MO in the absence of NAFLD. Therefore, NAFLD presence, but not obesity, was associated with high plasma levels of pro-NT. These results are similar to the studies of Barchetta et al. [13,14] and contradict our previous results that reported low levels of NT in women with MO and NAFLD [8]. These differences could be due to the molecule analyzed, because in our previous study [8], we analyzed NT levels in NAFLD for the first time. As some authors suggested later, pro-NT, a fragment precursor of NT, is a more stable molecule [15], which could explain these discrepancies.

In the current study, we analyzed pro-NT levels according to the hepatic histological classification, and we found significant differences between NL and NASH groups. However, we did not find relevant differences between SS and NASH as was reported in the study of Barchetta et al. [20]. This is probably due to the differential characteristics between populations, such as age, sex, and BMI.

Additionally, we could not find any association between levels of pro-NT and the presence of lobular inflammation or hepatic ballooning, as was reported in a recent study by Dongiovanni et al. [21]. These authors also described an interesting association between several NT genetic variants with advanced fibrosis and hepatocarcinoma in NAFLD patients, which likely affects NT protein activity.

With all this evidence, some authors considered pro-NT to be a peptide involved in the NAFLD/NASH pathogenesis through increased absorption of intestinal lipids and the induction of proinflammatory conditions in adipose tissue [15].

Finally, we determined whether pro-NT was related to lipid metabolism, given that NT can increase intestinal lipid absorption [13]. The only relationship we found was a positive correlation with LDLc levels. Our results were in agreement with Barchetta et al. study, where they also found a positive correlation between LDLc levels and pro-NT [13]. This association can be explained by different facts. First, our obese patients did not present severe dyslipidemia; only 15 of them presented mild dyslipidemia, and 9 of these patients were under statins treatment. Second, our MO patients underwent a very-low-calorie diet three months prior to the surgery, while our patients with NW followed a normal diet; thus, MO patients had lower levels of LDLc than NW subjects. Finally, in MO patients, as NAFLD progresses, pro-NT levels increase, promoting intestinal lipid absorption and triggering the increase in circulating LDLc [22].

Then, we analyzed the hepatic expression of some genes related to lipid metabolism. Accordingly, we found differential expression of SREBP2, ABCG1c, CROT, FAS, LXRα, PPARδ, and PPARγ in our cohort of patients with MO with or without NAFLD. In this sense, our NAFLD patients showed increased expression levels of SREBP2 and ABCG1, genes involved in lipoprotein secretion, and CROT and FAS, which are related to lipid oxidation and hepatic lipid accumulation [23,24]. Meanwhile, our NAFLD patients reported decreased levels of PPARδ and PPARγ, which were previously related to an increase in the insulin sensitivity [25]. When we evaluated the possible relationship of pro-NT levels with the hepatic expression of these lipid metabolism-related genes, we were unable to demonstrate any associations. In this sense, in the study of Barchetta et al. [14], plasma pro-NT levels were associated with high TG levels after adjustment for multiple confounders [20].

It is important to mention that although our studied population made it possible to establish a clear relationship between women with MO and NAFLD and the alteration of pro-NT levels, without the interference of confounding factors, such as sex or age, these results cannot be extrapolated to men, women of other ages, or individuals that are overweight or NW. Some patients in our cohort were on oral antidiabetic and lipid-lowering treatments, which are drugs with potential implications on the transport of bile that have been shown to affect the secretion of other intestinal peptides [26]. For this reason, the effect of these treatments on pro-NT levels was studied, but we did not find significant differences between patients with or without these therapies.

## 4. Materials and Methods

### 4.1. Subjects

The study was approved by the institutional review board (23c/2015), and all participants gave written informed consent. The study population was 74 Caucasian women: 18 controls with NW (BMI < 25 kg/m^2^) and 56 women with MO (BMI > 40 kg/m^2^). We conducted the study only in women to avoid interference from various confounders, such as gender. Liver biopsies were obtained from women with MO during planned laparoscopic bariatric surgery. All biopsies were indicated for clinical diagnostic purposes.

Diagnosis of NAFLD was made by using the following criteria: hepatic pathology and an intake of ethanol less than 10 g per day. The exclusion criteria for patients with MO were: (1) concurrent use of drugs that cause hepatic steatosis; (2) patients that take lipid-lowering medications, including peroxisome PPARα or PPARγ agonists; (3) diabetic patients receiving insulin or taking medications that were likely to influence endogenous insulin levels; (4) menopausal and postmenopausal patients or those patients taking contraceptive treatment; and (5) patients with an acute illness or current evidence of inflammatory diseases, acute or chronic infections, or end-stage malignant diseases.

In the cohort of subjects with MO, 16 women had T2DM, a diagnosis based on the guidelines of the American Diabetes Association, of which 88% presented antidiabetic treatment. Regarding women with NAFLD, 14 of them had T2DM. In addition, 15 women of the whole cohort presented dyslipidemia, of which 60% were receiving lipid-lowering treatment. Hypertension was present in 27 women of the whole cohort, of which 78% received antihypertensive treatment.

### 4.2. Sample Size

Accepting an α risk of 0.05 and a β risk of < 0.2 in a bilateral contrast, 20 participants per group were needed to detect a difference ≥ 0.2 units. It was assumed that the common standard deviation (SD) was 0.3.

### 4.3. Liver Pathology

Hepatic samples were scored by an experienced hepatopathologist using the methods described elsewhere [27,28]. Depending on their liver pathology, patients with MO were subclassified by using the following classification: NL histology (*n* = 20); SS (micro/macrovesicular steatosis without inflammation or fibrosis, *n* = 17); and NASH (*n* = 19).

### 4.4. Biochemical Analyses

A complete anthropometrical, biochemical, and physical examination was carried out on each patient. Biochemical parameters were analyzed using a conventional automated analyzer after 12 h fasting. IR was estimated using HOMA2-IR.

Pro-NT levels were measured using the sphingotest^®^ pro-NT assay (SphingoTec GmbH, Hennigsdorf, Germany). This is a chemiluminescence sandwich immunoassay using two monoclonal antibodies directed against pro-NT (tracer and capture antibody). The assay was calibrated using dilutions of native pro-NT. Samples/calibrators (50 μL) were pipetted into white polystyrene coated 96-well microtiter plates (Greiner Bio-One International AG, Kremsmünster, Austria). After adding labeled anti-pro-NT mAb (200 μL), the microtiter plates were incubated for 20 h at 22 °C without agitation. Unbound tracer was removed using washing solution (350 μL per well, four times), and remaining chemiluminescence was measured for 1 s per well using the Centro LB 960 microtiter plate luminescence reader (Berthold Technologies GmbH & Co. KG, Bad Wildbad, Germany). The pro-NT level was determined using a five-point calibration curve (5.9–482 pmol/L). Calibrators and samples were run in duplicate with a required < 15% coefficient of variation between the duplicates. The mean value of duplicates of each sample was reported. The analytical sensitivity (limit of detection) was <3 pmol/L, and the measurable range was 5.9–482 pmol/L.

### 4.5. Gene Expression in the Liver

Hepatic samples collected during bariatric surgery were conserved in RNAlater (Qiagen, Hilden, Germany) at 4 °C and then processed and stored at −80 °C. Total RNA was extracted from tissues by using the RNeasy mini kit (Qiagen, Barcelona, Spain). Reverse transcription to cDNA was performed with the High-Capacity RNA-to-cDNA Kit (Applied Biosystems, Madrid, Spain). Real-time quantitative PCR was performed with the TaqMan Assay predesigned by Applied Biosystems for the detection of SREBP1c, SREBP2, ABCG1c, ABCA1, ACC1, CPT1A, CROT, PPARα, PPARδ, PPARγ, FAS, LXRα, FXR, and FABP. The expression of each gene was calculated relative to the expression of 18S RNA. All reactions were carried in duplicate in 96-well plates using the 7900HT Fast Real-Time PCR system (Applied Biosystem, Foster City, CA, USA).

### 4.6. Statistical Analysis

The data were analyzed using the SPSS/PC+ for Windows statistical package (version 23.0; SPSS, Chicago, IL, USA). The Kolmogorov–Smirnov test was used to assess the distribution of variables. Continuous variables were reported as the mean (standard deviation); non-continuous variables were reported as the median and the interquartile range. The different comparative analyses were performed using a nonparametric Mann–Whitney U test or Kruskal–Wallis test, according to the presence of two or more groups. The strength of the association between variables was calculated using Spearman’s method. *p*-values < 0.05 were statistically significant.

## 5. Conclusions

In conclusion, our study confirmed that pro-NT levels are elevated in NAFLD/NASH and associated with LDLc levels. Therefore, although further studies are necessary to confirm the pro-NT role in NAFLD progression and to evaluate this molecule as a possible NASH therapeutic target, it does seem to be related to the development of this prevalent disease.

## Figures and Tables

**Figure 1 metabolites-11-00373-f001:**
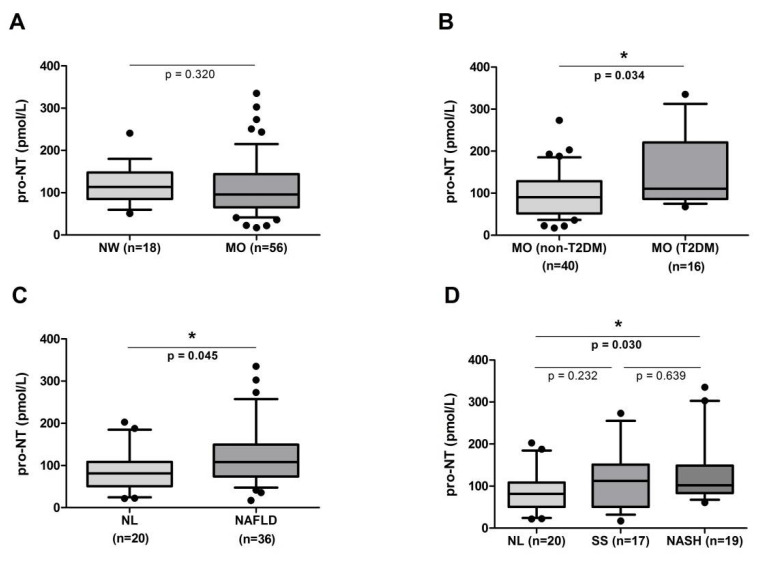
(**A**) Serum levels of pro-NT in women with NW and women with MO. (**B**) Serum levels of pro-NT in diabetic women with MO and non-diabetic women with MO. (**C**) Serum levels of pro-NT in women with MO and NL vs. NAFLD. (**D**) Serum levels of pro-NT in women with MO and NL vs. SS vs. NASH. NW, normal weight; MO, morbid obesity; NL; normal liver; NAFLD, nonalcoholic fatty liver disease; SS, simple steatosis; NASH, nonalcoholic steatohepatitis. * *p* < 0.05 is considered statistically significant.

**Figure 2 metabolites-11-00373-f002:**
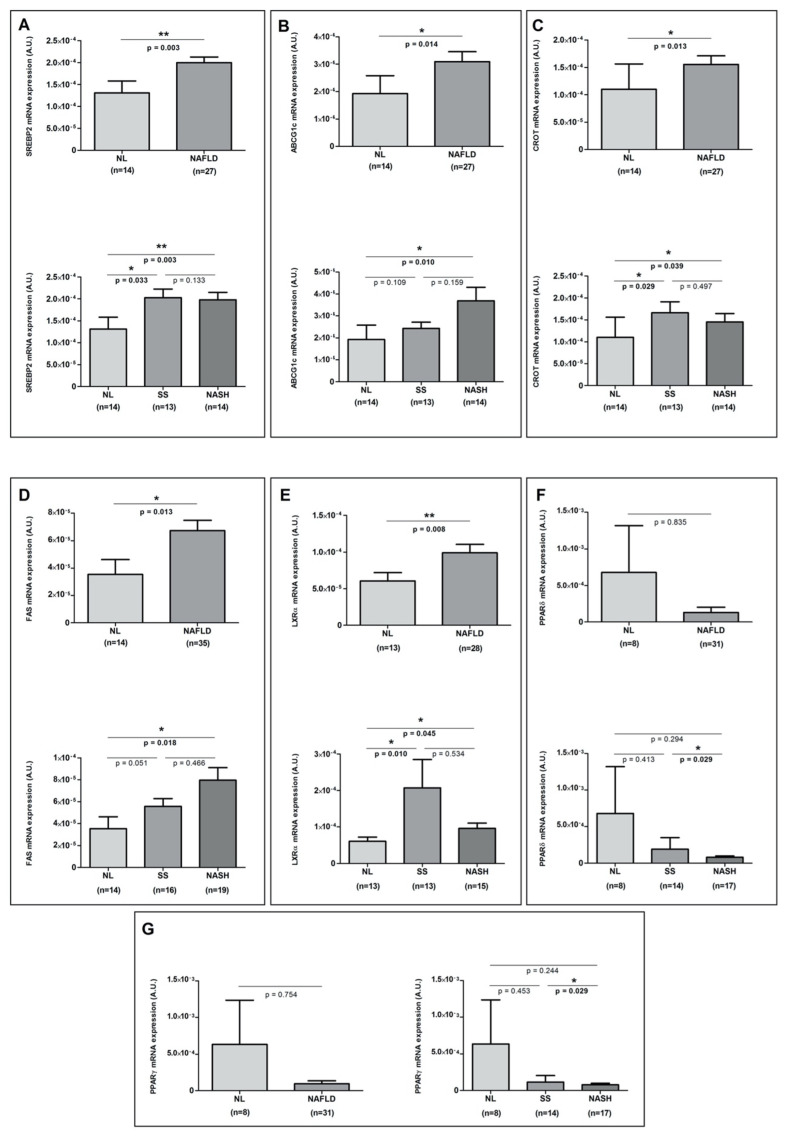
Differential mRNA hepatic expression of (**A**) SREBP2, (**B**) ABCG1c, (**C**) CROT, (**D**) FAS, (**E**) LXRα, (**F**) PPARδ, and (**G**) PPARγ between NL vs. NAFLD groups and between NL, SS, and NASH groups. NL; normal liver; NAFLD, nonalcoholic fatty liver disease; SS, simple steatosis; NASH, nonalcoholic steatohepatitis; SREBP2, sterol regulatory element-binding protein 2; ABCG1c, ATP-binding cassette (ABC) transporter 1c; CROT, carnitine O-octanoyltransferase; FAS, fatty acid synthase; LXRα, liver X receptor alpha; PPARδ, peroxisome proliferator-activated receptor delta and PPARγ, peroxisome proliferator-activated receptor gamma. * *p* < 0.05 and ** *p* < 0.01 are considered statistically significant.

**Figure 3 metabolites-11-00373-f003:**
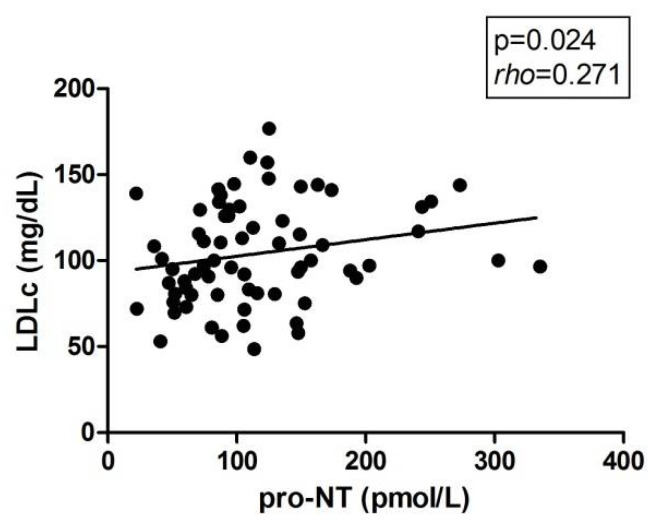
Correlation between plasma pro-NT and LDLc levels. Pro-NT, pro-neurotensin; LDLc, low-density lipoprotein cholesterol. *p* < 0.05 is considered statistically significant.

**Table 1 metabolites-11-00373-t001:** Anthropometric and biochemical variables of the study classified according to the BMI and the hepatic histopathological classification.

Variables	NW (*n* = 18)	MO (*n* = 56)		NAFLD (*n* = 36)	
		NL (*n* = 20)	NAFLD (*n* = 36)	SS (*n* = 17)	NASH (*n* = 19)
Age (years)	43.50 ± 6.73	44.54 ± 10.02	48 ± 9.79	46.19 ± 10.78	49.63 ± 8.78
Weight (kg)	54.2 (51–65.43) ^a^	115.5 (41.54–50.79)	119.5 (109–129.75)	120 (109.80–134)	119 (107–123)
BMI (kg/m^2^)	21.97 (20.79–24.08) ^a^	46.41 (41.53–50.79)	46.5 (44.03–51.69)	46.87 (43.03–56.09)	46.2 (44.26–48.59)
WC (cm)	71.5 (68.5–82.5) ^a^	125.25 (115–144)	130 (124–136.5)	133 (124–137)	129 (122.75–133)
Glucose (mg/dL)	90 (84.5–98.50) ^a^	83 (76–95)	116 (103–152) ^b^	115.5 (101.50–139.25) ^c^	116 (103–152) ^d^
Insulin (mUI/L)	7.8 (4.90–10.06) ^a^	11 (7.31–14.03)	16.31 (10.69–24.26) ^b^	17.6 (10.60–25.4)	15.24 (10.78–22.50) ^d^
HOMA2-IR	1.05 (0.60–1.30) ^a^	1.15 (0.90–1.65)	2.3 (1.45–3.4) ^b^	2.65 (1.30–3.45)	2 (1.50–3.20) ^d^
HbA1c (%)	5 (4.6–5.3) ^a^	5 (4.6–5.3)	5.5 (5–6.5) ^b^	5.8 (5–6.2) ^c^	5.1 (4.9–6.6)
Cholesterol (mg/dL)	197.59 ± 30.21 ^a^	166.60 ± 29.64	181.14 ± 34.12	177.25 ± 36.29	184.42 ± 32.81
HDLc (mg/dL)	64 (49.75–73) ^a^	48 (40–57)	38.10 (35.75–43.25) ^b^	36.5(29–41.05) ^c^	41 (36.75–44) ^d^
LDLc (mg/dL)	114.46 ± 28.85	89.61 ± 25.27	107.01 ± 29.12 ^b^	105.49 ± 30.69	108.53 ± 28.38 ^d^
Triglycerides (mg/dL)	85 (52.5–169.25) ^a^	124 (75–167)	162 (122.50–239) ^b^	168 (109.37- 243.75)	160 (124–239)
AST (UI/L)	19.5 (15.25–22.50) ^a^	21 (18.25–26.25)	32 (24.75–54) ^b^	31.5 (25.25–54) ^c^	36.5 (20.50–52.50) ^d^
ALT (UI/L)	15 (11.50–20.50) ^a^	18.5 (16–27.25)	35 (27–53) ^b^	33 (27.50–51.25) ^c^	37 (24–62) ^d^
GGT (UI/L)	11 (9–21) ^a^	17 (10.50–26)	27.5 (15.75–43) ^b^	30 (16–41)	25 (15–66)
ALP (Ul/L)	55.71 ± 14.67 ^a^	60.33 ± 11.49	70.80 ± 15.71 ^b^	68.87± 15.58 ^c^	72.51 ± 16.07 ^d^

^1^ NW, normal weight; MO, morbid obesity; NAFLD, nonalcoholic fatty liver disease; NL, normal liver; SS, simple steatosis; NASH, nonalcoholic steatohepatitis; BMI, body mass index; WC, waist circumference; HOMA2-IR, homeostatic model assessment method 2 of insulin resistance; HbA1c, glycosylated hemoglobin; HDLc, high-density lipoprotein cholesterol; LDLc, low-density lipoprotein cholesterol; AST, aspartate aminotransferase; ALT, alanine aminotransferase; GGT, gamma-glutamyltransferase; ALP, alkaline phosphatase. Data are expressed as the mean ± standard deviation or median (interquartile range), depending on the distribution of the variables. ^a^ Significant differences between NW subjects and patients with MO (*p* < 0.05). ^b^ Significant differences between patients with NL and NAFLD (*p* < 0.05). ^c^ Significant differences between patients with NL and SS (*p* < 0.05). ^d^ Significant differences between patients with NL and NASH (*p* < 0.05).

## Data Availability

The data presented in this study are contained within the article.

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
