# Peer review of "Circulating Levels of Pro-Neurotensin and Its Relationship with Nonalcoholic Steatohepatitis and Hepatic Lipid Metabolism"

_metabolites, 2021, doi:10.3390/metabo11060373_

Round 1
Reviewer 1 Report
Beatriz Villar and co-authors performed a study indicating the existence of a correlation between the level of circulating pro-neurotensin (pro-NT) and NAFLD-NASH.
The work is well written, and the results are presented in a clear manner.
There are numerous papers reporting that one of the causes of NAFLD is the augmented lipogenesis, supported by increased levels of lipogenic enzyme expression, analyzed by RT-qPCR and by Western blotting. In contrast, the authors reported that hepatic expression of the main lipid metabolism-related genes was found to be deregulated in NAFLD (Abstract). How the authors explain this discrepancy between their results and other data in the literature?
It is known that changes in mRNA levels do not always correspond to similar changes in protein levels. Maybe an in-depth analysis of the variations in gene expression had to be carried out through experiments to quantify the levels of protein, rather than mRNA. This aspect deserves to be discussed.
Overall, the work is interesting and provides additional evidence that could indicate pro-NT as a molecule likely involved in the etiology of NAFLD.
Reviewer 2 Report
The paper analyzes the relationship between circulating levels of pro-neurotensin (pro-NT) and nonalcoholic steatohepatitis and hepatic lipid metabolism in women.
The results show that pro-NT levels are significantly higher in women with morbid obesity without nonalcoholic fatty liver disease. Additionally, pro-NT levels are higher in patients with nonalcoholic steatohepatitis than in patients with normal liver. The hepatic expression of the main lipid metabolism-related genes is deregulated in nonalcoholic fatty liver disease. Additionally, although pro-NT levels correlate with LDL, there are no association with main lipid metabolism-related genes. Authors suggest that pro-NT could be related to NAFLD progression.
This referee has some concerns:
- Authors declare they analyzed only women to avoid gender bias. It should be better to also compare males.
- The analyzed women are in fertile age. Authors should stratify the data considering the different phases of woman hormonal period.
- Authors should analyze also the hepatic hydroxy methylglutaryl Coenzyme A reductase (HMGCR) and Low density Lipoprotein receptor (LDLr) which are the key players of cholesterol metabolism.
Reviewer 3 Report
The paper: “Circulating levels of pro-neurotensin and its relationship with nonalcoholic steatohepatitis and hepatic lipid metabolism” by Villar et al. showed higher pro-neurotensin levels in plasma of female patients with NASH. Plasma-neurotensin was associated with LDL cholesterol but not with mRNA expression of lipid metabolism genes in the liver.
Authors analysed pro-neurotensin in a cohort where neurotensin was already analysed before. Cohorts did, however, differ somehow. Number of patients was not exactly the same. Why?
MO patients had relatively low LDL. Why? Only 7 patients were on statin therapy.
Neurotensin has a very short half-life (about 30 s in mice). The pro- neurotensin is a stable precursor fragment and is produced in equimolar amounts relative to NT. Authors showed higher levels of pro-neurotensin in NASH patients in this study. In their previous work they noticed a decline. Do authors suggest that NL is more rapidly degraded in NAFLD? Current analysis confirms previous data published by Barchetta et al. but did not dissolve the disagreement of the authors´ current and previous study.
“The main novelty of the present study is that it clarifies previous discrepancies in relation to neurotensin in NAFLD and investigates its relationship with lipid metabolism.” This sentence is not correct. If analysis of neurotensin levels was correct in the previous study, it is unclear why there was a decline in NASH.
“Postprandial plasma NT levels have also been shown to be increased after gastric bypass surgery, suggesting that the regulation of NT secretion is altered in human obesity [11,12].” Reference 11 refers to pro-neurotensin.
Hepatic expression of pro-neurotensin was not analysed? Why? Dongiovanni et al reported that gene expression was correlated with fibrosis markers.
Why do the authors expect an association of fasting pro-neurotensin levels in serum and hepatic expression of e.g. FAS?
How do they explain that fasting pro-neurotensin levels are associated with LDL-cholesterol?
Is there an association of fasting and postprandial pro-neurotensin?
Round 2
Reviewer 2 Report
I do not have any further suggestion.
Reviewer 3 Report
authors have adressed my previous concerns.